# Care bundle to reduce readmission in patients with heart failure: a modified Delphi consensus panel in Argentina

Javier Roberti ,[1,2] Tomás Vita,[2] Jimena Piastrella,[3] Carlos Porley,[3] Lisandro Pereyra,[3] Mirta Diez,[4] Florencia Renedo,[5] Enrique Fairman,[6] Alberto Fernández,[7] Jorge Thierer,[8] Ezequiel García Elorrio[2]

► Prepublication history and supplemental material for this paper are available online. To view these files, please visit the journal online (http://dx.doi.org/10.1136/bmjopen-2020-040028).

For numbered affiliations see end of article.

**Correspondence to**
Mr Javier Roberti;
jroberti@iecs.org.ar

## ABSTRACT

**Objectives** The aim of this study was to develop consensus among Argentine cardiologists on a care bundle to reduce readmissions of patients with heart failure (HF).

**Setting** Hospitals and cardiology clinics in Argentina that provide in-hospital care for patients with HF.

**Participants** Twenty-four cardiology experts participated in the two online rounds and 18 (75%) of them participated in the third-round meeting.

**Methods** This study used a mixed-method design; it was conducted between August 2019 and January 2020. The development of a care bundle (a set of evidence-based interventions applied to improve clinical outcomes) involved three phases: (1) a literature review to define the list of interventions to be evaluated; (2) a modified Delphi panel to select interventions for the bundle and (3) definition of the HF care bundle. Also, the process included three rounds of scoring.

**Results** Twenty-six interventions were evaluated. The interventions in the final bundle covered four categories: medication, continuum of care, lifestyle habits, predischarge tests. These were: medication: beta-blockers, angiotensin receptor neprilysin inhibitors or ACE-inhibitors, furosemide and antimineralocorticoids; continuum of care: follow-up appointment, daily weight monitoring; lifestyle habits: smoking cessation counselling and low-sodium diet; predischarge tests: renal function, ionogram, blood pressure control, echocardiogram and determination of decompensating cause.

**Conclusion** Following a systematic mixed-method approach, we have developed a care bundle of interventions that could decrease readmission of patients with HF. The application of this bundle could contribute to scale evidence-based interventions.

## INTRODUCTION

Despite several medical advances to treat heart failure (HF), mortality and hospital readmission have not changed significantly.[1] The adherence to treatment and other related responsibilities demanded by the health system place a significant burden on patients and their caregivers.[2 3] Moreover, a high percentage of patients with HF are not receiving an adequate treatment despite the

### Strengths and limitations of this study

► Potential interventions were chosen through a systematic review.
► Cardiologist experts participated in a transparent consensus process.
► As in most consensuses, participants could have misinterpreted statements.
► Potential bias from cardiologists as only specialty involved in process.

increased use of both evidence-based therapies and performance measures.[4–6] In this context, the use of a care bundle with additional strategies such as quality improvement collaboratives (QICs) to scale up its use could contribute to the optimisation of the treatment of patients with HF.[7 8]

A care bundle is defined as a set of evidence-based interventions, called elements, which should be applied together in every eligible patient to enhance the reliability of care and to improve clinical outcomes.[9 10] The completion of the interventions of a bundle should be measured as all or nothing; when all components were performed collectively and reliably, they improved patient outcomes.[11] Therefore, a care bundle approach for HF should focus on providing evidence-based clinical practice, engaging patients and careers as active partners, and creating processes to ensure a quality handoff from hospital care.

Hospitalisations in HF are the main trigger for treatment interventions; however, financers' lack of awareness of the clinical burden and the low urgency to intervene in these patients compared with other cardiovascular diseases represent significant challenges for implementation.[12 13] In fact, HF guidelines recommend initiating and uptitrating disease-modifying therapies during hospitalisation.[14] As the preliminary phase

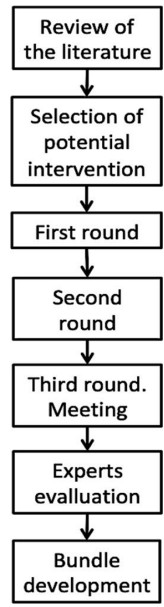

**Figure 1** Flow chart of Delphi process.

of a future QIC, the aim of this study was to develop consensus among Argentine cardiologists on a care bundle to reduce hospital readmissions of patients with HF.

## METHODS

This study used a mixed-method design and was conducted between August 2019 and January 2020. The approach used to develop a care bundle involved three phases: (1) a literature review to define the list of interventions that would be evaluated; (2) a modified Delphi panel to select interventions for the bundle; finally, (3) development of the final HF care bundle. The process included seven steps, with three rounds of scoring. See figure 1 for an illustration of the study design.

### Phase 1
#### Step 1: explorative review of the literature
In preparation of the Delphi questionnaire to be distributed, a review of the literature was performed, using a pragmatic exploratory approach. We searched in PubMed, LILACS, EMBASE, The Cochrane Library and Google Scholar for relevant literature. No HF care bundle was identified in the literature; although a review of grey literature showed isolated experiences shared over the internet. We included articles describing interventions for HF, with special attention to those specified in the Guidelines of the American College of Cardiology (ACC), the American Heart Association, the European Society of Cardiology (ESC) and the Argentine Cardiology Society.[7 8 15] Online supplemental annex 1 shows search strategies used. Online supplemental annex 2 shows the Preferred Reporting Items for Systematic Reviews and Meta-Analyses flow chart of selection process.

### Step 2: developing the list of interventions
The target patient for the interventions was an individual with signs and symptoms of HF and an ejection fraction lower than 40% who had been admitted to hospital and was expected to be discharged during the following 48 hours. Based on the literature, we proposed a list of 26 interventions to be assessed by experts. To guide the selection of interventions during the consensus process, we used the criteria proposed by GRADE from Evidence to Decision (EtD) framework validated and adopted by WHO (DECIDE project) and others from the care package literature to help panels of experts use evidence in a structured and transparent way to inform decisions in the context of healthcare recommendations.[10] Based on the literature, we selected a list of 26 interventions to be assessed by experts. Box 1 shows the 26 interventions. The potential interventions were presented to panellists as short statements with references to supporting literature. For each statement, panellists were asked to rate interventions according to 13 criteria or parameters using a nine-point Likert scale ranging from 1 (extremely inappropriate / insignificant) to 9 (extremely appropriate/ significant).[16] The 13 criteria were (1) priority, (2) unintended effects, (3) intended effects, (4) balance between unintended and intended effects, (5) quality of evidence on effects, (6) values and preferences, (7) resource required, (8) quality of evidence on resources, (9) cost-effectiveness, (10) equity, (11) acceptability, (12) feasibility and (13) measurability. The EtD framework ensures that panels consider important criteria, contributes to structure discussions and to identify the reasons for disagreements, based on the balance of health benefits and harms, human rights and sociocultural acceptability, health equity, equality and non-discrimination, societal implications, financial and economic considerations, feasibility and health system considerations and the meta-criterion quality of evidence.[17 18]

### Phase 2: consensus on a care bundle
#### Step 1: Selection of experts' panel
We used a modified RAND/UCLA Appropriateness Method in which experts used their professional judgement alongside the best available evidence to identify areas where consensus could be reached for the topic under consideration.[16] A total of 26 experts in chronic HF were selected from different clinical contexts in Argentina, mainly from the high-density area of Buenos Aires. All participants were specialists in cardiology, with proven experience in treating patients with HF and in scientific research on HF, were currently practising in hospitals which had a cardiology fellowship, at least 80 beds, and with 3000 admissions or more per year. These physicians were known to the team for their involvement in HF treatment and research and some of them were recommended by the first selected participants. The 26 potential participants were sent a recruitment letter and information sheet via email; follow-up telephone calls were made to confirm that the material had been received. Of the total,

**Box 1    List of interventions proposed**

Potential interventions
1. ß-blockers at least at initial dose, plan increase to maximum tolerable dose during outpatient care.
2. ACE-inhibitors or angiotensin II antagonists at least at initial dose, plan increase to its maximum tolerable dose during outpatient care.
3. Mineralocorticoid antagonists at initial dose, plan increase to maximum tolerable dose if ejection fraction is ≦ 35% during outpatient care.
4. Ivabradine at least at initial dose and plan increase to maximum tolerable dose during outpatient care, if heart rate (HR) >70 bpm despite ß-blockers at maximum tolerable dose and being in sinus rhythm.
5. Angiotensin receptor-neprilysin inhibitor at least at initial dose, plan increase to maximum tolerable dose during outpatient care.
6. Diuretic dose should be adjusted based on imaging, laboratory, and functional class examinations.
7. Implant a resynchroniser in case of complete blockage of the left branch with a QRS>0.15 s, in sinus rhythm, receiving optimal medical treatment and persisting in heart failure class III–IV.
8. Antipneucoccal and influenza vaccines.
9. Measure ferritin and correct if low (<100 ng/mL or 100 to 300 ng/mL if transferrin saturation is <20%).
10. Measure urea, creatinine, calculate glomerular filtration rate (GFR) and study ionogram (sodium, potassium and chlorine) to establish prognosis and therapy feasibility.
11. Perform cardiac ultrasound and communicate results to treating physician to establish therapeutic action, adjust medication and improve risk stratification.
12. Perform pulmonary ultrasound and communicate result to treating physician to establish therapeutic action, adjust medication and improve the patient's risk stratification.
13. Fit pulmonary artery pressure monitoring device for treating physician to improve monitoring and adjust medication.
14. Measure B-type natriuretic peptide (BNP) and N-terminal pro B-type natriuretic peptide (NT-proBNP) and report to treating physician to establish therapeutic action, adjust medication and improve the patient's risk stratification.
15. Monitor blood pressure and try to reach values between 130/80 and 110/70, prioritising use of ß-blockers and ACE-inhibitor or angiotensin receptor neprilysin inhibitor.
16. Instruct patient to weigh daily and consult or increase the diuretic dose if increase is ≧ 2kg kg in 3 days.
17. Indicate patient to ingest a maximum of 2 g of sodium per day and inform which foods are rich in sodium.
18. If patient is a smoker, recommend smoking cessation and an action plan to achieve it.
19. Advise patient to lose weight if obese or overweight and not in advanced stages of congestive heart failure (CHF).
20. Advise patient to perform aerobic exercise at least four times per week for 40 min min up to 80% of predicted HR for their age.
21. Within 1 week of discharge, specialist nurse should call patient to follow up on adherence to dietary hygiene measures, medication and its adverse effects, weight gain and monitoring by the cardiologist.
22. Schedule an appointment for a consultation with the specialist within 1 week of discharge.
23. Patient should be evaluated by a psychologist/psychiatrist to screen for depression and initiate treatment accordingly.

Continued

**Box 1    Continued**

24. Reassure patient that there are no restrictions on sexual activity. If phosphodiesterase inhibitors are used, the patient should be informed about contraindications and precautions.
25. Identify the cause of decompensation.
26. If patient has obstructive sleep apnoea, treatment with continuous positive airway pressure should be initiated.

24 (92%) professionals accepted the invitation. Panellists were given a small honorarium for their time and effort.

### Step 2: interventions rating

Permanent disagreement was defined as at least six panellists rating the indication in the 1–3 region, and at least six panellists rating it in the 7–9 region after three rounds of scoring. Each intervention obtained 13 scores, one for each parameter or criterion.[16] Then, this score was multiplied by the weighting factor obtained in the rating of the parameters and mean was calculated for each intervention. Interventions were then sorted according to this global score.

### Step 3: first round online survey

For the first round, the experts received a link for the online questionnaire (surveymonkey.com, San Mateo, California, USA). Twenty-six interventions in 13 criteria were evaluated for disagreement following the classic RAND definition of at least six panellists rating the parameter in the 1–3 region and at least six panellists rating it in the 7–9 region.[16] Security sockets layer encryption was used to protect data while being transmitted by ensuring secure connections between participants and the server. An email was issued to each participant requesting them to open the survey tool, answer the questions and press the submit button. All participants had the contact details for the research team that could be contacted in case of any doubts regarding the materials provided. Participants had 10 days to complete the survey; a reminder email was sent to non-responders.

### Step 4: second online survey round

In round 2, all interventions and criteria that had not reach an agreement in the first round were included for a new vote. Participants were shown the whole group's ratings as frequency data on a rating scale for each item, alongside their own ratings for each statement in round 1. They could revise their own original rating considering the group ratings if they so wished. Participants had 10 days to complete round 2; a reminder was sent to non-responders.

### Step 5: consensus meeting

All participants of the online survey were invited via email and telephone to a 3-hour in-person consensus meeting. The main objectives of the meeting were to discuss the indications that have been rated with disagreement and inconsistencies in appropriateness ratings. Panellists

were encouraged to refer to the literature. Of the total of participants who responded the online rounds, 18 (75%) took part in the meeting. During the session, results of the second online survey were presented and discussed, followed by an anonymous onsite rating of those items that had showed disagreement on previous rounds. In addition, the 13 quality criteria were anonymously rated using a 1–9 Likert scale to obtain a factor for final evaluation of each intervention. The panel mean score was calculated for each statement and classified into three categories: 1–3.5 (potentially inappropriate); 3.6–6.4 (uncertain) and 6.5–9 (potentially appropriate). The moderator was a physician, not a specialist in the area to avoid bringing their bias to the discussion, had led the literature review and was familiar with the material.

### Step 6: development of HF care bundle with expert committee

We invited five of the participants of previous rounds with extensive expertise to join an expert committee to discuss the results of the survey and rate the bundle. The results of the process were presented in this meeting. The expert committee selected those interventions with the highest scores for the final evaluation and no disagreements. Based on these results, a final bundle of interventions was developed.

### Ethical aspects and role of funding source

Written informed consent was obtained from all participants included in the study, and all procedures were conducted according to the Declaration of Helsinki. The founder of the study did not intervene in any stage of the consensus process.

### Patient and public involvement

The research question addresses a most significant issue for patients and was informed by a review of the literature on patients' journey when admitted for HF care. Only physicians with a vast clinical experience participated in the consensus process; these participants brought patients' concerns and perspectives into the process. The results of this study are disseminated via open-access publication. This study is complemented by a qualitative research on patients' perspectives, reported separately.

## RESULTS

Twenty-four experts participated in the two online rounds and 18 (75%) of them participated in the third-round meeting. Eight (33.3%) were female; 16 (66.7%) participants worked in the private sector, 2 (8.3%) in the public sector and 6 (25%) in both sectors. Sixteen (66.7%) participants worked in the metropolitan area of Buenos Aires, while the rest, 8 (33.3%) participants, worked in other provinces of Argentina. The median of professional experience was 26.5 years (range 9–40 years).

Twenty-six interventions were evaluated, and each intervention obtained a global score. After the three rounds, consensus was reached for all interventions except for one

intervention under one criterion (unintended effects of telephone tracking of patients). The mean global scores of 13 interventions were above the cut-off score of 6.5 and 7 of these reached a median score of 7.1 (online supplemental table 1). Of the 13 final interventions, five (38%) were related to medications, two (15%) were diagnostic tests and two (15%) were habit recommendations ('hyposodic diet' and 'daily weight monitoring'). The two most valuable criteria ranked by experts were the priority and the desirable effects of each intervention. 'Administration of ACE-inhibitors', the 'evaluation of renal function', 'performance of an ionogram' and 'blood pressure control' were considered the most important interventions. The intervention 'use of beta-blockers' was ranked in the fifth position while the 'use of antimineralocorticoids', 'furosemide' and 'angiotensin receptor neprilysin inhibitor (ARNI)' were in the 9th, the 10th and the 11th positions, respectively. The use of ARNI positioned better on undesirable effects than the administration of furosemide and antimineralocorticoids. 'The administration of ARNI' and 'the use of ACE-inhibitors' obtained the same scores in six criteria; mainly related to scientific evidence evaluation; of note, under the criteria evaluating equity, ACE-inhibitors obtained a higher score. 'Smoking cessation' and 'blood pressure control' were ranked higher than the use of medication because, unlike medication, the first two had no potential undesirable effects according to participants.

'Daily weight monitoring' and 'a follow-up appointment within a week of discharge' were ranked in the fourth and the seventh positions, respectively. 'Echocardiography during hospitalisation' and 'determining the cause of decompensation' were ranked above the established cut-off point. 'Measurement of biomarkers B-type natriuretic peptide (BNP) and N-terminal pro B-type natriuretic peptide (NT-proBNP)' ranked below the cut-off point. 'Physical exercise' and 'cardiac resynchronisation therapy' did not rank highly for the overall goal of reducing hospital readmission.

### Proposed bundle

The proposed bundle was based on four categories of interventions: medication, continuum of care, lifestyle habits, predischarge tests (table 1). Beta-blockers, ARNI or ACE-inhibitors, furosemide and antimineralocorticoids were included as interventions related to pharmacological treatment. ARNI or ACE-inhibitors can be used interchangeably. The election would depend on clinician judge and patient choice. Follow-up appointment with a specialist within a week of discharge and daily weight monitoring were also included as interventions that reduce 30-day hospital readmission under 'continuum of care' category, while smoking cessation counselling and low sodium diet were interventions classified as lifestyle habits. The interventions included under pre discharge tests were 'renal function evaluation', 'performance of ionogram', 'blood pressure control', 'performance of an

**Table 1** Proposed bundle

| Interventions | |
|---|---|
| Predischarge medications | Beta-blockers-reach optimal dose |
| | ACE-inhibitors or angiotensin receptor/neprilysin inhibitor reach optimal dose |
| | Antimineralocorticoids-reach optimal dose |
| | Furosemide-reach optimal dose |
| Continuum of care after discharge | Follow-up appointment with the specialist 7 days after discharge |
| | Daily weight monitoring under same conditions |
| Lifestyle habits | Low sodium diet-intake of <2 g of sodium per day |
| | Tobacco cessation counselling |
| Predischarge tests | Renal function and ionogram evaluation |
| | Blood pressure control (predischarge) |
| | Echocardiogram during hospitalisation |
| | Determination of the decompensating cause |

echocardiogram during hospitalisation' and 'determination of the decompensating cause'.

## DISCUSSION

Following a validated and systematic consensus method, we created a care bundle of interventions that could contribute to the management of HF and prevent readmission within the 30 days after discharge. The reduction of readmissions of patients with HF represents a challenge because several factors such as a lack of a targeted therapy for patients with preserved ejection fraction, the natural history of HF, the need for daily management to avoid decompensations and problems related to access to care, health literacy and other socioeconomic factors.[19] HF is a complex syndrome and the result of congestion and/or poor perfusion, leading to negative effects on multiple organ systems. In fact, patients with HF have multiple comorbidities, a heavy medication burden and experience symptoms that may be attributed to several aetiologies.[19]

Several studies have evaluated single interventions to improve outcomes in patients with HF. Although the ESC, ACC and HF Argentinean guidelines have established recommendations; critical interventions have not been specified.[7 8 15] For participants, the priority and desirable effects were the two most important criteria to evaluate each intervention. Probably, these two criteria are what cardiologists in the local context refer to when assessing any treatment in time-constrained settings. We have defined priorities in a time constrained setting as it is usually the case with clinical work. As all the potential interventions had been recommended on HF guidelines, no significant differences in global scores were observed among those above the cut-off point. Indeed, when the

results were shown to the experts, they all agreed on the significance of the interventions that were finally included in the bundle.

Almost half of the interventions were related to medication, which, with robust evidence supporting its efficacy, is widely accepted as the backbone of HF treatment. It has been shown that interventions to improve adherence to medications for HF significantly reduce the risk for both hospitalisation and death.[4] Interestingly, the use of beta-blockers was only fifth on the interventions ranking although these drugs have been confirmed to reduce mortality, readmission and improve symptomatology.[20–23] It is possible that the use of beta-blockers was not chosen as the most important intervention in our study as many patients have HF in very advanced stages and require inotropes during hospitalisation, while beta-blockers are administered during outpatient care. Clinical trials of ARNI showed a significant reduction in mortality and readmission rates when compared with ACE-inhibitors.[24 25] The administration of ACE-inhibitors was pondered as the most important intervention and, at a granular level, the difference between ARNI and ACE-inhibitors was based mainly on costs and access. The high cost of the drug in an economically constrained setting explains the gap between evidence and real-world practice. Due to new evidence, the final recommendations in the bundle allow the physician to decide between ARNI or ACE-inhibitors.[24 25]

Interestingly, many of the interventions were not strictly clinical practices such as daily weight monitoring or a follow-up appointment, which reflects the importance of an integrated care and the complex setting in which professionals and the health system interact to achieve the best quality outcomes. Moreover, in a follow-up appointment, the patient could be provided with information to prevent readmission. Importantly, laboratory examinations such as tests of renal function and the ionogram obtained a high score and were valued as the second most important intervention. Probably related to this, of the five medications included in the bundle, four could negatively affect renal function. Also, in the Argentinian health system, BNP test is expensive and not covered by most health insurances, which could explain the fact that this test was not included in the final list of elements.

In pioneering care bundles, two components were essential to their success; participating clinicians agreed that there was sufficient medical evidence supporting each individual element.[26–29] Second, the list of elements in the bundle should not be extensive; the goal of the bundle approach is to develop a short list of interventions already recommended and accepted in guidelines. This allows improvement rather than a debate on validity of the elements. Moreover, it is essential that elements have the consensus of local clinicians.[10] On the other hand, checklists and protocols serve to augment memory and limit the chance of human error, improving communication, standardising responses and reducing unnecessary clinical variation.[30]

Our study has limitations that should be acknowledged. Participants could have misinterpreted the original statements describing the potential interventions and the same applies to criteria definitions. We tried to control this factor by providing participants with information and discussing definitions during the meeting. An important limitation is related to the method itself; our results reflect the opinion of selected specialists with a vast experience. Cardiologists see most of HF patients and once the bundle applicability is assessed, it could be disseminated to other contexts that include other specialties. The interventions were not tested in community or primary care, but this bundle is intended to be used during patient hospitalisation, before discharge. Also, we disclosed the funding by a pharmaceutical company and the limited participation of professionals affiliated to this company, which could be interpreted as a limitation. We have made every effort to maintain the transparency of the process in every stage.

In conclusion, following a mixed-method approach, we developed a care bundle of interventions that could decrease readmission of patients with HF. Our next step will be to study the implementation of this bundle in several sites. The application of this bundle could be used to scale evidence-based interventions under the framework of Quality Improvement Learning Collaborative.

**Author affiliations**
[1]CIESP, National Scientific and Technical Research Council (CONICET), Buenos Aires, Argentina
[2]Health Care Quality and Patient Safety, Institute for Clinical Effectiveness and Health Policy, Buenos Aires, Argentina
[3]Heart Failure, Novartis Argentina SA, Buenos Aires, Argentina
[4]Heart Failure Service, Instituto Cardiovascular de Buenos Aires, Buenos Aires, Argentina
[5]Heart Failure Service, Fundacion Favaloro Hospital Universitario, Buenos Aires, Federal District, Argentina
[6]Heart Failure Service, Clinica Bazterrica, Buenos Aires, Argentina
[7]Cardiology Service, Sanatorio Modelo Quilmes, Quilmes, Argentina
[8]Heart Failure Service, CEMIC, Buenos Aires, Argentina

**Acknowledgements** We thank the participation of the following health professionals: José Luis Barisani (Clínica Adventista), Roberto Colque (Sanatorio Allende), Marcelo Crespo (Sanatorio Los Arcos), Clara Huerta (Hospital Córdoba), Juan Pablo Escalante (Instituto Cardiovascular Rosario), David Flores (Hospital Nacional de Clínicas), Jimena Gambarte (Hospital Alemán), Daniela García Brasca (Hospital Italiano de Córdoba), Alfredo Larraburu (Sanatorio Los Arcos), Adrián Lescano (Centro Gallego), Javier Marino (ICBA), Lidia Lobo Márquez (Instituto de Cardiología), Cristian Mastantuono (Hospital Pirovano), Felipe Martínez (Universidad Nacional de Córdoba), Stella Pereiro (Hospital Churruca), Eduardo Perna (Instituto de Cardiología de Corrientes Juana Francisca Cabral), José Santucci (Hospital Austral), Julieta Soricetti (Hospital Durand), Ezequiel Zaidel (Sanatorio Güemes).

**Contributors** JR: methodology, investigation, analysis, writing. TV: investigation, analysis, writing. JP, CP and LP: investigation, interpretation, review. MD, FR, EF, AF and JT: investigation, interpretation, review. EGE: concept, methodology, analysis, interpretation, review.

**Funding** This study was supported by Novartis Argentina SA (grant number NCC CLCZ696BAR05R).

**Competing interests** JP, CP and LP are employees of Novartis Argentina SA, funder of this study. JR, TV, FR, EF, AF, JT and EGE have no competing interest to declare.

**Patient consent for publication** Not required.

**Ethics approval** This study was evaluated and approved by an independent ethics committee, Independent Ethics Committee for Clinical Pharmacology Trials Professor Luis M. Zieher, Buenos Aires, Argentina, protocol number 0042/19 V.05 on 17 October 2019.

**Provenance and peer review** Not commissioned; externally peer reviewed.

**Data availability statement** Data are available on reasonable request.

**ORCID iD**
Javier Roberti http://orcid.org/0000-0002-4285-5061

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
