## [Reviewer comments · BMJ Open]

ARTICLE DETAILS

TITLE (PROVISIONAL)	Care Bundle to Reduce Readmission in Patients with Heart Failure: A Modified Delphi Consensus Panel in Argentina
AUTHORS	Roberti, Javier; Vita, Tomas; Piastrella, Jimena; Porley, Carlos; Pereyra, Lisandro; Diez, Mirta; Renedo, Florencia; Fairman, Enrique; Fernandez, Alberto; Thierer, Jorge; Garcia Elorrio, Ezequiel

VERSION 1 – REVIEW

REVIEWER	Peter Lachman ISQua Ireland/UK I know final author
REVIEW RETURNED	24-May-2020

GENERAL COMMENTS	An interesting paper to develop a care bundle in Argentina. From a methodological point of view there no concerns on the conduct of the modified Delphi process, except to state that it is the consensus of cardiologists and heart failure is not managed only by cardiologists so there is that bias to address. A number of queries to be clarified - How do you define expertise - e.g. the final 5 had 'extensive' expertise - what does that mean and could that imply a bias?- on the interventions some are outside the domain of the cardiologists - were these concepts and interventions tested with those in community or primary care ?- did you test whether the care bundle can be implemented i.e. is it practical or not and whether it worked - if used in a collaborative would it work?- did you consider the prevention aspects in great detail and how these could be implemented?- did you compare this care bundle with other HF care bundles? - I give one for example from other countries - give one for example from Scotland https://www.heartfailurehubscotland.co.uk/models-of-care/ there are also ones from Wales England etc. This would assist you in deciding whether the bundle you have developed has comparative validity Is this the first step for the collaborative. If so I recommend that the bundle be tested in one site to see if it works, and can be implemented and which other professionals on the care pathway - from person to primary care to secondary care to cardiologist - should be involved. Basically if you have the cardiologists look at the whole pathway without others on the pathway the bundle will have a bias to their view rather than a wider view.
---

VERSION 1 – AUTHOR RESPONSE

Reviewer's comment

1. Comment: How do you define expertise - e.g. the final 5 had 'extensive' expertise - what does that mean and could that imply a bias?

Response: We thank the reviewer for your comment. All participants were specialists in cardiology, needed to have ample experience in treating patients with HF and in scientific research on HF, practiced in hospitals which had a cardiology fellowship, at least 80 beds, and with 3,000 admissions or more per year. These physicians were known to the team for their involvement in HF treatment and research and were recommended by first participants to be selected. These physicians were known to the research team for their involvement in HF research and were also recommended by first participants to be selected. Then, to draft the final document, a core group of experts was selected from the 26 participants, based on years of experience.

2. Comment: On the interventions some are outside the domain of the cardiologists - were these concepts and interventions tested with those in community or primary care?

Response: We thank the reviewer for this comment. All the interventions evaluated before the bundle selection have a level of evidence IA-IIB in ACC/AHA, ESC and Argentinian guidelines to be performed in admitted patients with HF. These interventions can be performed by professionals outside the cardiology area, interventions are simple and clear. We have added the reviewer's point as a limitation of this study.

3. Comment: Did you test whether the care bundle can be implemented i.e. is it practical or not and whether it worked if used in a collaborative would it work?

Response: We thank the reviewer for the opportunity to clarify this important point. At this first phase, this paper describes the process of developing the bundle through a Delphi method. We then conducted a formative research with rapid qualitative methods to assess professionals' perceptions and opinions on the bundle, potential barriers and facilitators. These findings informed the following stage of the study which is the implementation of the bundle in 10 sites. We hope we will be able to publish our results to show if the bundle could be implemented, and how it worked. We have included this information in the final paragraph of the discussion.

4. Comment: Did you consider the prevention aspects in great detail and how these could be implemented?

Response: We thank the reviewer for this comment. We agree, prevention aspects are important, and they were discussed in the group meetings. Some of the interventions were related to giving patients tools that could help them prevent other events in the future. We believe that the inclusion of the intervention "Follow up appointment with the specialist seven days after discharge" intends to cover prevention education. We wanted to make sure through this intervention that the patient is discharged with a secured appointment for the following week, and that in this appointment, the physician has enough quality time to provide the patient with information on prevention. We have now included this idea in the discussion. However, prevention aspects were not further discussed because this bundle is to be implemented during hospitalization, predischarge.

5. Comment: Did you compare this care bundle with other HF care bundles? - I give one for example from other countries - give one for example from Scotland

<https://www.heartfailurehubscotland.co.uk/models-of-care/>

there are also ones from Wales England etc. This would assist you in deciding whether the bundle you have developed has comparative validity.

Response: We thank the reviewer for this suggestion. The resources provided are of great help. All the bundles provided have comparable to the final bundle we have developed; this is not surprising

because all interventions have a level of evidence of IA-IIB. In fact, to compare and learn from other HF bundle initiatives, we ran a systematically search in PubMed, LILACS, EMBASE, The Cochrane Library and Google Scholar relevant literature and we could not find any HF bundle published in peer review journals. We have added a comment on this point in the discussion.

6. Comment: Is this the first step for the collaborative. If so, I recommend that the bundle be tested in one site to see if it works and can be implemented and which other professionals on the care pathway - from person to primary care to secondary care to cardiologist - should be involved.

Response: We thank the reviewer for this recommendation. As we explained in a previous point, this paper reports the development process, but when have the results of the implementation study we would like to publish those results, probably including findings from the formative research. Thank you for your recommendation. In this step we only aimed to determine a useful bundle. In our case, selected sites differed quite substantially, so, piloting the bundle in one site may not be applicable to the others. The collaborative is intended to interact and improve the bundle implementation taking into account the different actors of each site.

7. Comment: Basically if you have the cardiologists look at the whole pathway without others on the pathway the bundle will have a bias to their view rather than a wider view.

Response: We thank the reviewer for pointing this issue. We have assessed the pathway according to local clinical idiosyncrasy, to the local context. Moreover, we have included this point as a limitation.

VERSION 2 – REVIEW

REVIEWER	Peter Lachman ISQua
REVIEW RETURNED	21-Nov-2020
GENERAL COMMENTS	Thank you for the revision of the paper and addressing the points made in the first review. The next step will be to test the use and impact of the bundle.

VERSION 2 – AUTHOR RESPONSE

Thank your for your prompt reply on the revised version of our manuscript. We have checked the paper and made corrections as necessary. Your suggestions were all implemented and additional corrections were made. These are highlighted in the marked document. A clean copy was also included. Please find detailed responses below:

There are still various typographical/ grammatical errors in the paper. Can you please thoroughly proofread the manuscript one mroe time? Some examples are included below but please note this is not an exhaustive list.

Comment. Page 3: “As in most consensus, participants could have misinterpreted statements” Do you mean something like: “As in most *consensuses*..”?

We thank the editors for pointing out this mistake. We have corrected this sentence.

Comment. Page 14: “many of the interventions were not strictly clinical issue” should be “many of the interventions were not strictly clinical *issues*”

Thank you. We have rewritten this phrase.

Comment. Page 14: “Importantly, laboratory examinations such as tests of renal function and the ionogram, obtained a high score and valued as the second most important intervention.”

We thank the editor for pointing out this error. We have made changes in this sentence.

Comment: Page 3: Why is “Consensus was reached through Delphi process” a strength or a limitation? This appears to be just describing what happened. Please remove or revise this point. It should be clear why each point in this section is a strength or limitation relating to the study’s design and/ or methods.

We thank the editor for this comment. We have deleted this sentence from the list.

Some examples are included below but please note this is not an exhaustive list.

We have made several small changes throughout the text, which are highlighted in the marked copy, to correct these mistakes and improve the manuscript readability.